# Modulation of Immune Infiltration of Ovarian Cancer Tumor Microenvironment by Specific Subpopulations of Fibroblasts

**DOI:** 10.3390/cancers12113184

**Published:** 2020-10-29

**Authors:** Ji Wang, Frank H. C. Cheng, Jessica Tedrow, Wennan Chang, Chi Zhang, Anirban K. Mitra

**Affiliations:** 1Indiana University School of Medicine-Bloomington, Indiana University, Bloomington, IN 47405, USA; jwa7@iu.edu (J.W.); frcheng@iu.edu (F.H.C.C.); colemjee@iu.edu (J.T.); 2Medical and Molecular Genetics Department, Indiana University School of Medicine, Indianapolis, IN 46202, USA; wnchang@iu.edu; 3Department of Electrical and Computer Engineering, Purdue University, West Lafayette, IN 47907, USA; 4Melvin and Bren Simon Comprehensive Cancer Center, Indiana University, Indianapolis, IN 46202, USA

**Keywords:** ovarian cancer, tumor microenvironment, cancer-associated fibroblasts, immune cells, immunotherapy, metastasis

## Abstract

**Simple Summary:**

The ovarian cancer tumor microenvironment is made up of ovarian cancer cells along with a milieu of proteins and normal cells, including fibroblasts, immune cells, endothelial cells, pericytes and adipocytes. The noncancer components also play an important role in determining the fate of the tumor and exhibit a lot of heterogeneity. In this study, we have used a deconvolution algorithm to identify four different fibroblast subpopulations and multiple immune cell types, from bulk RNA-seq data of ovarian cancer primary tumors, metastases and normal omentum. We report the prevalence of specific fibroblast subtypes that determine the tumor-immune microenvironment. Our study can potentially help provide a template for identification of potential combination therapies to enhance the efficacy of ovarian cancer immunotherapies.

**Abstract:**

Tumor immune infiltration plays a key role in the progression of solid tumors, including ovarian cancer, and immunotherapies are rapidly emerging as effective treatment modalities. However, the role of cancer-associated fibroblasts (CAFs), a predominant stromal constituent, in determining the tumor-immune microenvironment and modulating efficacy of immunotherapies remains poorly understood. We have conducted an extensive bioinformatic analysis of our and other publicly available ovarian cancer datasets (GSE137237, GSE132289 and GSE71340), to determine the correlation of fibroblast subtypes within the tumor microenvironment (TME) with the characteristics of tumor-immune infiltration. We identified (1) four functional modules of CAFs in ovarian cancer that are associated with the TME and metastasis of ovarian cancer, (2) immune-suppressive function of the collagen 1,3,5-expressing CAFs in primary ovarian cancer and omental metastases, and (3) consistent positive correlations between the functional modules of CAFs with anti-immune response genes and negative correlation with pro-immune response genes. Our study identifies a specific fibroblast subtype, fibroblast functional module (FFM)2, in the ovarian cancer tumor microenvironment that can potentially modulate a tumor-promoting immune microenvironment, which may be detrimental toward the effectiveness of ovarian cancer immunotherapies.

## 1. Introduction

Ovarian cancer (OC) is the most lethal gynecologic malignancy and is the fifth leading cause of cancer-related deaths among women in the USA [1]. Most OC patients are diagnosed at a late stage because the symptoms are confused with other common ailments which, as a result, leads to poor prognoses [2,3,4]. In spite of responding to the initial cytoreductive surgery and carbo-taxol-based therapy, most OC patients are likely to relapse and eventually become chemo-resistant [5]. The patients with recurrent disease are treated with additional cycles of chemotherapy that may provide transient respite, but, ultimately, this is not curative. Tumor immunotherapies, such as checkpoint inhibitors, have been found to be effective in a constantly expanding variety of cancers, but only about 10% of OC patients have had objective responses in clinical trials [6,7]. A better understanding of tumor-immune responses in the context of other microenvironmental components may help to improve the outcome of immunotherapies in OC.

The tumor-immune infiltration plays an important role in the development of cancer progression [8]. Inflammation has been linked to carcinogenesis as evidenced by previous reports demonstrating that the immune infiltration could lead to tumor proliferation, metastasis, and angiogenesis in many different types of cancer [9,10,11]. In general, the inflammatory process plays an important role in tissue repair and healing [12]. However, the inflammation also enhances cellular proliferation and neovascularization, leading to development of carcinogenesis [13,14]. Tumor-infiltrating immune cells consist of many distinct cell types, such as natural killer (NK) cells, natural killer T-cells (NKT) cells and cytotoxic T lymphocytes (CTLs) [15]. NK cells have been demonstrated to be regulators of angiogenesis through producing several angiogenic factors, including vascular endothelial growth factor (VEGF), placental growth factor (PGF) and interleukin 8 (IL-8) [16,17]. NKT cells produce interleukin 13 (IL-13) through the interleukin 4 receptor (IL-4R)-signal transducer and activator of transcription 6 (STAT6) pathway, which causes myeloid cells to secrete transforming growth factor beta (TGF-β) and promote metastasis [15,18,19,20]. Another report has suggested that CTLs may promote the cancer cells to produce prostaglandin E2 (PGE2) and increase the recruitment of myeloid-derived suppressor cells (MDSCs) through the Fas-signaling pathway, thus indicating that CTLs are involved in the process of immunosuppression [21]. On the other hand, tumor-associated macrophages (TAMs) are predominant in the inflammation-mediated processes, and there is extensive evidence of the association of high TAM density within metastasis in several types of cancers, including breast, colorectal, ovarian carcinoma and melanoma [13,22,23,24]. It is believed that antitumor immunity also arises from TAM-mediated immunosuppressive microenvironments, such as those possessing immunosuppressive cytokines [25].

The tumor microenvironment (TME) has been reported to provide favorable conditions for carcinogenesis [26]. Cancer-associated fibroblasts (CAFs) are critical members in the family of TME [27]. Several studies have demonstrated that CAFs could promote tumor progression through secretion of paracrine factors, production of cytokines or growth factors, promotion of angiogenesis and remodeling of the extracellular matrix (ECM) [28,29,30]. However, CAFs are a heterogenous population in which the subpopulations can have diverse functions [31,32,33]. Interestingly, CAFs have also been found to recruit TAMs through the secretion of interleukin 6 (IL-6), CC-chemokine ligand 2 (CCL2) and macrophage colony-stimulating factor 1 (M-CSF) [34,35,36,37]. Notably, CAFs have been demonstrated to activate the janus kinase 2-signal transducer-STAT3-CCL2-signaling pathway of the TME and to initiate immunosuppression by recruiting the regulatory T-cells (T_regs_) and circulating MDSCs [35,38]. A notable hypothesis is that CAFs may recruit many tumor-infiltrating immune cells, driving the OC to become more malignant [37]. Targeting CAF-mediated immunosuppression may even potentially sensitize tumors to immunotherapies [39,40].

To explore the role of CAFs and their varied functions in OC progression and metastasis, we conducted an extensive bioinformatic analysis of our and other publicly available ovarian cancer datasets (GSE137237: https://www.ncbi.nlm.nih.gov/geo/query/acc.cgi?acc=GSE137237; GSE132289: https://www.ncbi.nlm.nih.gov/geo/query/acc.cgi?acc=GSE132289 and GSE71340: https://www.ncbi.nlm.nih.gov/geo/query/acc.cgi?acc=GSE71340), by mining the correlation of fibroblasts within the TME with the characteristics of tumor-immune infiltration. Our analysis identified (1) four functional modules of CAFs in ovarian cancer that are associated with the TME and metastasis of OC, (2) immune-suppressive function of the collagen 1,3,5-expressing CAFs in primary OC and omental metastases, and (3) consistent positive correlations between the functional modules of CAFs with anti-immune response genes and negative correlation with pro-immune response genes.

## 2. Results

### 2.1. Study Design

To gain an understanding of the role of the fibroblasts in the TME and their influence on tumor-immune response, we compared the fibroblast populations and immune profiles of primary tumors vs. matched metastasis and also metastatic tumors vs. normal omentum. We analyzed our RNA-seq data, comparing the transcriptomes of primary tumors and matched metastasis from ovarian cancer patients [41]. Similarly, we analyzed the RNA-seq data of normal omentum and omental metastasis in patients and syngeneic mouse models [42,43].

Figure 1 illustrates the computational analysis framework. CAF and immune cell and sub-cell types, cell type-specific functions and related markers were systematically compared among different conditions in the collected data. Noting CAFs and other immune cell types in ovarian cancer can consist of discrete subpopulations that affect the patient outcome in various ways [31],we applied our in-house cell subtype identification and deconvolution method, namely inference of cell types and deconvolution (ICTD, https://github.com/zcslab/ICTD), to accurately assess cell subtypes and predict their relative populations [44]. ICTD is specifically developed to identify dataset-specific gene markers of cell subtypes and their specific functions in bulk tumor sequencing data by a semi-supervised approach (see more details in Section 4).

In our previous study, we identified four subpopulations of fibroblasts from The Cancer Genome Atlas (TCGA) pan-cancer data that we call fibroblast functional modules (FFM), namely, non-collagen (FFM1)-, collagen 1,3,5 (FFM2)-, collagen 4 (FFM3)- and collagen 6 (FFM4)-expressing modules, based on their specific gene expression signatures (Appendix A) [44]. Averaged expression level of the gene markers of the four FFM were utilized to quantify the relative activities in each sample. Similarly, relative proportions of immune cell types presented in the microenvironment were identified using dataset-specific gene signatures identified by ICTD (Appendix A). The gene co-expression correlation of each FFM subtype in TCGA ovarian cancer data is provided in Appendix A. The OC TCGA primary tumors were segregated into three distinct subgroups with respect to their FFM marker gene expression (Appendix A). Further analysis of the average expression levels of FFM1–4, in each of the molecular subtypes of OC in TCGA data using published marker genes [45], revealed that the mesenchymal and the differentiated groups have enrichment of fibroblasts (Appendix A). The FFM2 levels were specifically high in the mesenchymal subtype, which is associated with poor prognosis [45]. A survival analysis of the effect of the individual FFM genes on the overall survival of serous ovarian cancer patients was further performed (Appendix A). To confirm that the FFM signatures were truly enriched in fibroblasts in the TME, we analyzed single cell RNA-seq data from ovarian cancer metastases [46]. The average expression levels of the FFM1–4 markers in various cell types were determined and the fibroblasts had significantly higher expression of the FFM signatures than all other cells in the TME (Appendix A). The percentage of each FFM subgroup in the fibroblasts present in the TME of metastases of each patient was also analyzed (Appendix A). Most of the fibroblasts belonged to the FFM subtypes except for a small fraction in each tumor. Thereafter, we studied the variation of the FFMs and immune cells in the normal omentum as compared to primary tumors and metastases. This helped identify the changes in the fibroblastic and immune cell populations occurring in the omentum as the metastasis is established as well as between primary tumors and metastases as ovarian cancer progresses and spreads. Correlational analysis was then performed between the predicted activity level of FFMs and the proportion of immune subtypes in the metastatic tumors to identify the influence of the specific FFM subtypes on the tumor-immune microenvironment. Finally, we compared all three datasets to identify immune-related genes that are common or unique.

### 2.2. Fibroblasts and Immune Profiles of OC Metastasis Compared with Matched Primary Tumors

We started with analyzing the proportions of immune cells in ovarian cancer patient metastasis compared to matched primary tumors [41]. We observed a significant increase in the overall T-cell population in the metastasis along with increased cluster of differentiation 8 (CD 8) T-cells, monocytes and myeloid cell reactive oxygen species (ROS) production (*p* < 0.05 by Mann–Whitney test) (Figure 2A). All four FFM populations were found to be significantly higher in the metastatic tumors compared to the matched primary tumor (Figure 2B). This indicated increased desmoplasia occurring as ovarian cancer disseminates. We further confirmed our findings in another publicly available dataset [47], where FFM2 was significantly increased in the metastases compared to matched primary tumors, while the other FFMs displayed a non-statistically significant increase in metastases (Appendix A). Thereafter, we proceeded to specifically look for correlations between the four FFM subpopulations with tumor immunity-related genes in the metastatic tumors. Figure 2C shows the tumor immunity-related genes that are significantly associated with at least one FFM subpopulation (*p* < 0.05) (with complete statistics given in Appendix A). The genes were clustered (unsupervised) into immune-activating and immune-suppressing groups based on their functions. All the FFM groups positively correlated with the immune-suppressing genes and had a negative correlation with the immune-activating genes (Figure 2C, Appendix A). Taken together, this indicated that the increased desmoplasia in the metastatic tumors caused an inhibition of antitumor-immune responses while enhancing immune tolerance. A correlation analysis between cell types in the metastasis showed a similar trend in correlation of FFM1, 3 and 4 with the immune cells while FFM2 differed (Figure 2D). The former three negatively correlated with macrophages, dendritic cells and T-cells, while the latter had a negative correlation with the relative cytotoxicity of T-cells, MDSCs and neutrophils. Here, relative cytotoxicity of T-cells is defined by the log ratio of expression level of cytotoxic marker genes (perorin 1/granzymes) and total T-cell markers (CD2/CD3) [44].

### 2.3. Comparison of the Microenvironments of the Normal Omentum and Omental Metastasis

Having assessed the changes in the fibroblast and immune profiles during metastasis, we focused on the microenvironmental changes induced at the metastatic site. Cancer cells reaching the site of metastasis initiate a process of reprogramming the resident stromal cells into tumor-associated stroma [27,48,49]. This process leads to the conversion of resident normal fibroblasts into CAFs, generating tumor-immune tolerance in addition to reprogramming the immune cells to promote metastatic tumor growth. Therefore, we compared the immune cell profiles of normal omentum and omental metastases using publicly available RNA-seq data from multiple syngeneic mouse models of high-grade serous ovarian cancer [43]. Maniati et al. had characterized polyclonal high-grade serous ovarian cancer syngeneic mouse models (HGS), the original genetically engineered mouse model (GEMM) [50] that they were derived from through backcrossing with C57BL/6 mice [43], and cell lines 302000 and 60577 (cl30200 and cl60577) developed from GEMMs. While this GEMM model gave rise to high-grade serous ovarian cancer arising from the Fallopian tube secretory cells (as it occurs in the human disease [50]), another GEMM model was also studied that arose from the ovarian surface epithelium (unlike its human counterpart [51]). A principal component analysis revealed a clustering of the metastases from HGS and GEMM tumors (Appendix A), which was separated from the cl30200 and cl60577 cluster. Gene ontology (GO) analysis on the upregulated genes in the HGS/GEMM cluster vs. the cl cluster identified the upregulated genes are majorly associated with negative regulation of the immune process, indicating stronger immune responses in the HGS/GEMM cluster when comparing to the cl cluster (Appendix A). Therefore, we further restricted our analysis to the HGS and GEMM group. Compared to the normal mouse omentum, the metastatic tumors had significantly lower levels of total T-cells, monocytes, dendritic cells, MDSCs, B-cells and cytotoxic T-cells, while neutrophils were found at higher levels (*p* < 0.05 by Mann–Whitney test) (Figure 3A). While there was an increase in the total fibroblast population in metastasis compared to the normal omentum (*p* = 1.5 × 10^−12^), the adipocytes were significantly lower (*p* = 1.7 × 10^−11^) (Figure 3A and B). These opposing changes confirm previous reports that the ovarian cancer cells first use the fats stored in the adipocytes to fuel their growth and, once the adipocytes are depleted, the CAFs in the TME reprogram their metabolism toward using glycogen instead [52,53]. Among the fibroblast subpopulations, FFMs 2 (*p* = 1.8 × 10^−12^), 3 (*p* = 0.0074) and 4 (*p* = 1.5 × 10^−10^) increased significantly in the metastatic tumors, while FFM1 exhibited a nonsignificant decreasing trend (Figure 3B). A correlation analysis of the FFM groups with differentially expressed immune-related genes revealed a variability between the FFM groups (Figure 3C, Appendix A). While FFM2 and 4 had a strong positive and negative correlation with the immune-suppressive and immune-activating genes, respectively, FFM1 and 3 did not. Similarly, FFM2 and 4 were inversely correlated with most immune cells, indicating that these fibroblast subpopulations may be involved in tumor-immune escape (Figure 3D).

We next compared the omental metastasis with adjacent normal omentum using publicly available data from Fran Balkwill’s group (GSE71340) [42]. In humans, we found a significant increase in the regulatory T-cells (*p* = 0.01) and B-cells (*p* = 0.00021) in omental metastasis as compared to normal omentum (Figure 4A, Appendix A). To analyze the fibroblast subgroups, we correlated them with the disease score (Figure 4B) defined by the percentage of malignant cells in the tissue, the higher value of which indicates a greater presence of malignant cells in the tissue [42]. FFM2 had a significant positive correlation with the disease score (*R* = 0.53, *p* < 0.01), while FFM3 correlated negatively (*R* = −0.38, *p* < 0.05, Figure 4B). FFM1 and 4 did not correlate significantly with the disease score. This was further confirmed by the poor progression-free survival of ovarian cancer patients with tumors expressing high levels of the three collagens enriched in FFM2 (Appendix A). The analysis of correlation between the FFM types with immune-activating or immune-suppressing gene expression revealed a similar trend for FFM 2, 3 and 4. All of them correlated positively with immune-suppressing genes and negatively with the immune-promoting genes (Figure 4C, Appendix A). However, FFM1 appeared to be an exception. An analysis of the correlation between cell types revealed a similar trend of positive correlation between all four FFMs and immune cells (Figure 4D). Taken together, our analysis suggested that FFM2 is the main subgroup among microenvironmental fibroblasts that promote metastatic tumor progression and immune tolerance in human omental metastases.

### 2.4. Immune Gene Signatures of OC Metastasis

To identify the common immune-related genes that are deregulated during metastasis as compared to the primary tumors, as well as to the normal omentum, we compared the significantly deregulated genes from all three datasets (Figure 5A). Twelve genes were found to be common to all three comparisons that were potentially relevant to OC progression as well as reprogramming of the metastatic microenvironment. Further analysis of these 12 genes showed that the immunological functions of these genes generally correlated with the FFMs (Figure 5B). The six anti-inflammatory genes that had a positive correlation were predominated by the components of the TGF-β-signaling pathway. The proinflammatory genes negatively correlated with the FFMs. The two human datasets comparing primary tumors with matched metastases (GSE137237) and omentum metastases with adjacent normal omentum (GSE71340) had 34 differentially expressed immune-related genes in common (Figure 5A,C). The mouse metastasis vs. normal omentum data (GSE132289) shared 25 genes with the similar comparison in humans and 22 genes with the human primary tumors vs. metastasis (Figure 5A,C). Overall, our analysis suggested there was a positive correlation of the FFM subtypes with the anti-inflammatory genes and a negative correlation with the proinflammatory genes.

## 3. Discussion

The recent emergence of immunotherapy, including checkpoint inhibitors, to treat solid tumors is a promising new development for melanomas, lung cancer, head and neck cancers, etc. [54,55]. However, the clinical trials for immunotherapy in ovarian cancer have produced low response rates [56,57]. The balance of the pro- and antitumor-immune response is maintained by the genetic composition of the cancer cells as well as components of the tumor microenvironment and can have a strong influence on patient outcome [58]. Microenvironmental factors like special AT-rich sequence-binding protein 1, prostaglandin E2 and cyclooxygenase 2 can modulate the differentiation of dendritic cells in the TME into a tumor-tolerant phenotype [59,60]. OC stem cells can promote TME macrophages to polarize into the tumor-promoting M2 type [61]. CAFs constitute a major fraction of the tumor stroma in OC and produce various chemokines, cytokines and growth factors to promote tumor growth [27]. The potential of CAFs modulating the tumor-immune response has not been rigorously explored. Recent reports have established that CAFs are a heterogenous population capable of varying functions [62,63]. Subpopulations of CAFs have been reported to play a critical role in tumor-immune tolerance and prevention of effective immunotherapies in several cancers [63,64,65]. In this study, we conducted a tissue data deconvolution analysis of the RNA-seq data to OC data and established four broad subpopulations of fibroblasts, FFM1–4, in the TME of OC. It is important to take into consideration that these segregations are based on deconvolution of bulk RNA-seq data. Therefore, there is always a possibility that genes from these signatures are also expressed by other cell types in the TME. Our analysis of publicly available single cell RNA-seq data from ovarian cancer metastases demonstrated that the fibroblasts were significantly enriched for the FFM signatures, increasing the confidence in our deconvolution approach. Moreover, we observed versatile effects of the FFMs on different immune cell types as well as pro- and anti-inflammatory gene expression signatures.

Overall, our findings indicate that the FFM2 subtype, with high expression of collagen 1,3,5, plays a key role in tumor-immune tolerance and results in poor prognosis. The metastatic mouse ovarian cancer tumors had an inverse correlation of the FFM2 subtype with T-cells, myeloid cells, B-cells, dendritic cells, monocytes and neutrophils. The FFM2 was the only subtype showing a positive correlation with disease score in human metastasis and it correlated negatively with T-cell cytotoxicity, while all the fibroblasts were inversely correlated with regulatory T-cells. T-cell cytotoxicity was determined as the ratio of tissue cytotoxicity (expression of perforin 1 and granzyme A) to T-cells (expression of CD3 and CD2). Furthermore, we compared our immune cell type data with multiplexed immunohistochemistry immune profiling of high grade serous ovarian cancer (HGSOC) chemotherapy responders and poor responders [47]. Interestingly, increased T-cell populations in metastasis were consistent with increased T-cell infiltration observed in good responders to Carbo/Taxol chemotherapy. However, the increased B-cell population in omental metastasis we observed corresponded with poor outcome of chemotherapy.

Inflammation plays an interesting dual role in cancer progression. It can act in a manner that can be detrimental to the tumor or in a way that actually initiates or even promotes tumor development. T-cell-mediated cytotoxicity can lead to tumor inhibition [66] and, in ovarian cancer patients, its infiltration into tumors is generally associated with better prognoses [67]. Yet, it can also release various chemokines and proteases which can aid tumor development. Macrophages, similarly, can either kill early malignancies [68], or they can become TAMs, which function to provide growth factors and aid in angiogenesis for the tumor [69]. Inflammasomes can eliminate cancerous precursors via apoptosis or they can allow for tumor progression and interaction with the TME [66,68]. Chemokines can allow lymphocytes to migrate for development and to home in on various targets, yet they also can be used to enable metastatic spread. Different transcription factors, such as STAT3 [66] or nuclear factor-κB (NF-κB) [70], can amplify the immune cell population or they can be manipulated by the TME to allow for tumor proliferation and circumvention of apoptosis. The dichotomous nature of inflammation and its relationship with cancers provides important information for clinicians to determine cancer prognoses for their patients and to explain many of the paraneoplastic symptoms they experience [66]. It also provides a multitude of possible targets for future therapies as well as possible insights about how metastases might adapt to or circumvent them. Of the 12 shared inflammation-related genes in the three datasets, the six proinflammatory genes were inversely correlated with the fibroblasts, while the six anti-inflammatory ones had a positive correlation with them. This indicated the potential role of the fibroblasts in modulating inflammation in the TME.

Of the six anti-inflammatory genes, SMAD5 is expressed in macrophages [71], B-cells [72] and Bone marrow-derived mesenchymal stem cell (BM-MSCs) [73]. It helps in the polarization of TAMs into the M2 phenotype [71], leading to immunosuppression within the local tumor environment. Aryl hydrocarbon receptor (AHR), expressed in T helper 17 (Th17) cells [74], NK cells, macrophages [75] and dendritic cells (DC) [76], suppresses both the innate and adaptive immune responses. It leads to IL-10 production via NK cells [77] and macrophages [75], while also leading to production of T_regs_ [78]. Additionally, AHR leads to production of adenosine within the tumor microenvironment [78], ultimately leading to the blunting of T-cell, NK cell and macrophage function [79]. Transforming growth factor beta receptor 2 (TGFBR2), found on NK cells [80] and T-cells [81], have both proapoptotic and growth-suppressive functions [82], in addition to being able to suppress effector function of both NK [80] and T-cell populations [83]. TGFBR2′s suppression of innate immunity stretches beyond just the peripheral nervous system as well; it is also able to inhibit lipopolysaccharide-induced neuroinflammation in the brain [84]. In later cancer stages, TGFBR2 even functions to promote epithelial–mesenchymal transition (EMT) [80]. TGFB2 is secreted by macrophages [85] as well as by neural precursor cells (NPCs) [86], and has been shown to suppress the macrophage inflammatory response [87]; Transforming growth factor beta 3 (TGFB3), expressed by T-cells, was found to suppress B-cell immune responses and to lessen the production of antibodies [88]. Furthermore, it has been linked to the production of pathogenic Th17 cells [89]. TGF-β can cause immunosuppression by promoting inhibitory cell-cell contacts with effector T-cells in the ovarian cancer TME [90]. 5’-Nucleotidase Ecto (NT5E ) is expressed by T_regs_, anergic T-helper (T_h_) cells and epithelial cells. Like AHR, it produces the anti-inflammatory, proangiogenic molecule, adenosine.

Among the six proinflammatory genes, CD27, expressed in B-cells [91], T-cells and T_regs_, has been shown to enhance NK cell survival, B-cell activation and immunoglobulin G (IgG) production [92]. CD19, expressed by B-cells [91,93], is required for a multitude of B-cell functions, including survival, function, differentiation and chemotaxis [91]. While involved with humoral immunity, its signaling threshold also acts as a regulator of peripheral tolerance [94]. IL-21R is also primarily expressed by B-cells, though it is also seen in DC, NK cells and T-cells. It serves to enhance the innate immune response by alternatively activating macrophages, inducing apoptosis via DC, and amplifying NK cell function. It also boosts the adaptive immune response by enhancing T-cell effector functions, proliferation and survival, and through its role in plasma cell generation and germinal cell function [95]. CD80, expressed in DC, T-cells [96] and proinflammatory macrophages [97], functions to disrupt programmed cell death ligand 1/programmed cell death protein 1 (PD-L1/PD-1) binding. This, in turn, prevents the inhibition of T-cell activation [96]. In addition to this, it also regulates self-tolerance via T_reg_ homeostasis [98]. CCL5, expressed by macrophages, DC and T-cells, promotes tumor proliferation, metastasis and invasion [99]. It also is necessary for tumor-infiltrating lymphocyte (TIL) engraftment [67] and plays a role in the recruitment of macrophages [100] and inflammatory angiogenesis [101]. CCR7 is expressed by DCs, T-cells and B-cells [102].

Even though the presence of tumor-infiltrating lymphocytes suggests that these are immunoreactive tumors, clinical trials with PD-1 inhibitors have only resulted in modest response. The evidence of certain subpopulations of microenvironmental fibroblasts modulating inflammatory and tumor-immune profiles in OC TME presented in this study provide further evidence for the need to understand the role of the tumor stroma in determining the efficacy of OC immunotherapy. A recent single cell analysis in breast cancer revealed a positive feedback regulatory mechanism between T_regs_ and certain CAF subpopulations that caused immunotherapy resistance [63]. Our present study, using deconvolution methods and bulk sequencing data, advocates for similar single cell transcriptomic studies to further determine the key interactions and mechanisms of immune tolerance and potential therapy resistance in OC. At the same time, such deconvolution methods can serve as a cost-effective way to analyze clinical samples to determine if the patient is a good candidate for a specific immunotherapy.

## 4. Materials and Methods

### 4.1. Data Analyzed in This Study

Transcriptomic data of GSE132289, GSE71340 and GSE137237were retrieved from the NCBI GEO database (https://www.ncbi.nlm.nih.gov/geo/) and processed with standardized protocols. Specifically, GSE132289 data includes 37 mouse samples (16 HGS, 3 GEMM, 9 cl, 9 normal omental tissues), 8 human samples; GSE137237 data contains 11 pairs of matched primary and metastases tumor tissue samples; and GSE71340 data contains 35 samples. Deconvolution, coexpression correlation and differential expression analysis were separately conducted in each dataset.

### 4.2. Quantification of Cell Types and Cell Type-Specific Functions

We applied our in-house method, ICTD, to quantify the level of cell types and their functions. Specifically, ICTD identifies dataset-specific cell type and functional marker genes, which are further used to estimate relative proportion or activity level of the cell types or functions, by the following computational steps: (1) Construct a labeling matrix for cell type-specific gene expression in a given microenvironment. We first construct a labeling matrix to store the gene marker trained from bulk tissue data of selected cell types, fibroblast, adipocytes, endothelial cell, B-cell, CD4+ T-cell, CD8+ T-cell, natural killer cell, dendritic cell, monocytes, macrophages and neutrophil. The entry with higher value in the labeling matrix suggests the gene has a higher expression level in the cell type. (2) ICTD first detects dataset-specific cell marker genes by implementing a bi-cross validation (BCV) for rank test, and a nonparametric hub and module detection method [44]; this step generates lists of cell type marker genes for further deconvolution analysis. (3) Predict cell proportions using constrained Non-negative Matrix Factorization (NFM). With the marker genes identified, ICTD conducts the deconvolution by a constraint non-negative matrix factorization approach by minimizing, as in the following Equation (1):
(1)$$\underset{S,P}{\min}\left(\|X-S\cdot {P\|}_{F}^{2}+\lambda \cdot \mathrm{tr}\left(S\left(1-C^{T}\right)\right)\right)$$
where S, P and C represent matrices of gene signature, cell proportion and consistency with pre-predicted cell type-specific gene markers. (4) Estimate cell type-specific functions. For each cell type detected in (3), ICTD further screens to identify the marker genes of a varied cell type-specific function and utilizes a second round of NMF to estimate cell type-specific functions. See more details in https://github.com/zcslab/ICTD [44].

### 4.3. Differential Expression, Correlation and Pathway Enrichment Analysis

Differential gene expression analysis was conducted by using DESeq2 (v1.26.0), with *p* = 0.05 as the significant cutoff. Pearson correlation coefficients between genes and estimated cell proportions and functional module activity level were computed. Significance of the correlation was tested by Student’s *t*-test with *p* = 0.05 as the significant cutoff. Pathway enrichment analysis was conducted by utilizing Metascape (https://metascape.org/gp/index.html#/main/step1), with *p* = 0.05 as the significant cutoff.

### 4.4. Single Cell RNA-Seq Analysis

The raw gene expression count matrix of ovarian cancer patient 1 (OvC1) was downloaded from the E-MTAB-8107(https://www.ebi.ac.uk/arrayexpress/experiments/E-MTAB-8107/) dataset. The R Seurat package (v3.2.0) was utilized to analyze the count matrix, filtered by keeping cell barcodes (>400 unique molecular identifiers (UMIs)), expressed genes (>200 and <6000) and reads mapping to mitochondrial genes (<25%). Default normalization methods in the R Seurat package were applied for the matrix. By regressing out the number of UMIs, influences of cell cycle, percentage of mitochondrial genes and selecting 50 principal components (PCs), clusters were found by setting resolution to 0.5 within the FindClusters function in R Seurat. Annotation of clusters was completed by considering gene markers from each cluster. Cell clusters were categorized into 7 cell types (fibroblasts, endothelial cells, epithelial cells, erythroid cells, B-cells, myeloid cells and T-cells) and expression values for FFM1, FFM2, FFM3 and FFM4 were calculated by averaging the gene markers from them, respectively. Two-sided *t*-test was implemented to calculate the significance between fibroblast cell type and other cell types. Next, the fibroblast group was annotated and divided into 5 subgroups: FFM1, FFM2, FFM3, FFM4 and Others, using the FFM1–4 signatures. The “Others” group represents cells that were not assigned into the 4 FFM groups because of lack of expression of their markers.

## 5. Conclusions

Our study identifies a specific fibroblast subtype, FFM2, in the OC tumor microenvironment that can potentially modulate a protumorigenic immune microenvironment. Enrichment of the FFM2 fibroblast subpopulation may be detrimental toward the effectiveness of OC immunotherapies. Therefore, further studies focusing on the specific mechanism of action of these fibroblasts can help determine potential combination therapies that can enhance the efficacy of OC immunotherapies.

## Figures and Tables

**Figure 1 cancers-12-03184-f001:**
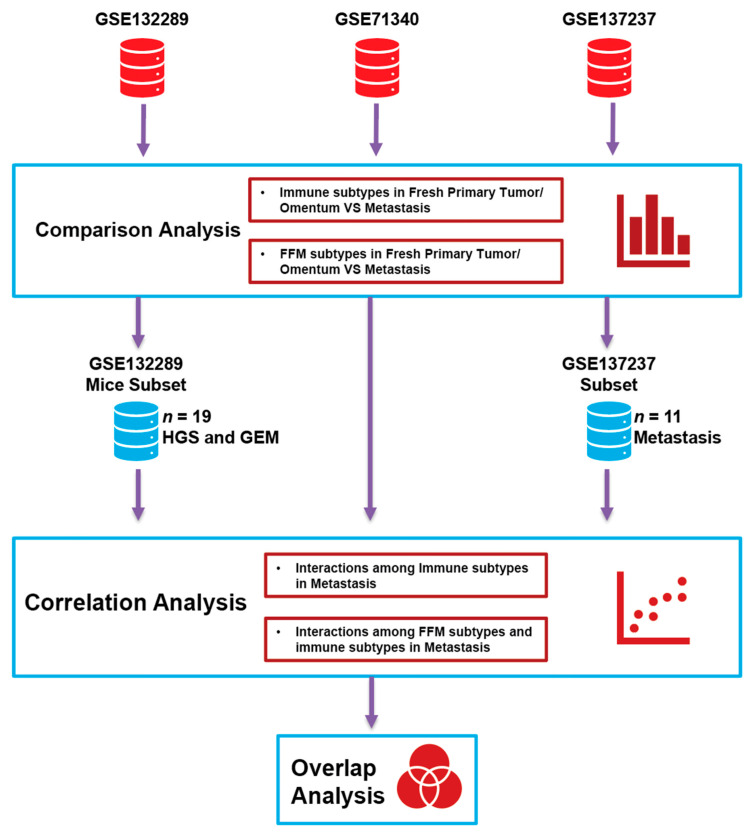
Analysis workflow: Three ovarian cancer (OC) RNA-seq datasets comparing normal omentum and omental metastasis (dataset GSE132289), omental metastasis (GSE71340) and matched primary tumors and metastasis (GSE137237) were analyzed. Only the normal omentum and omental metastasis from mice were included in the analysis from GSE132289. They were first analyzed for immune cell subtypes and fibroblast subtypes using a deconvolution algorithm developed in-house. Thereafter, correlations between the immune subtypes and fibroblast subtypes were determined to identify the role of specific microenvironmental fibroblasts in modulating the tumor-immune microenvironment. Finally, the immune-relevant genes common to the different datasets were identified. HGS: high-grade serous ovarian cancer syngeneic mouse models, GEM: genetically engineered mouse, FFM: fibroblast functional modules.

**Figure 2 cancers-12-03184-f002:**
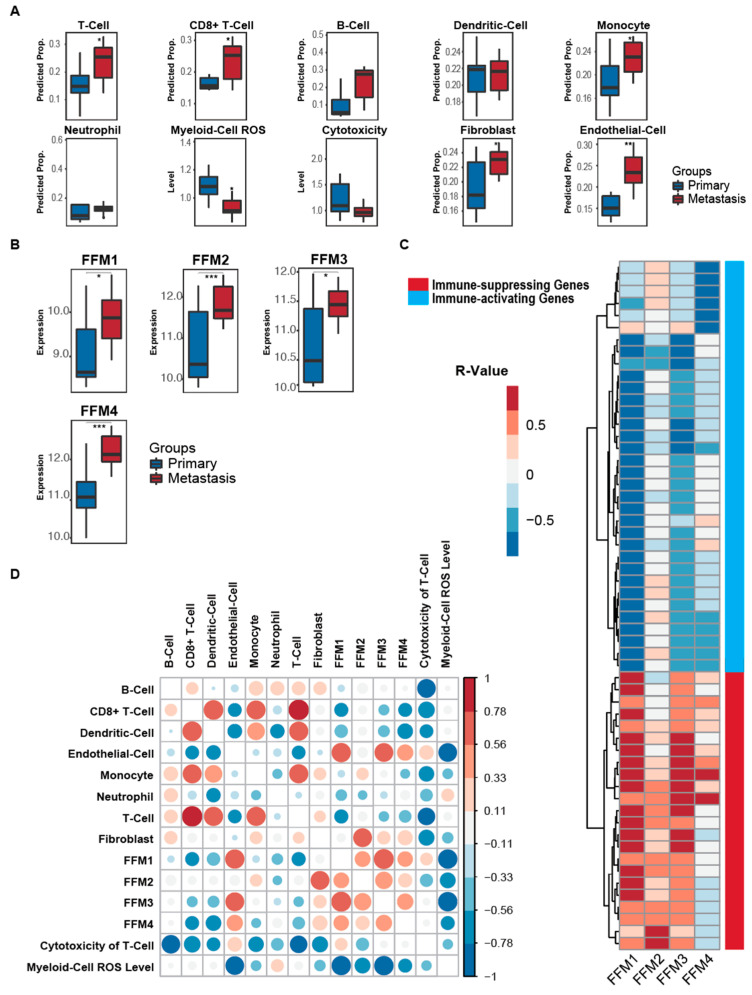
Characterization of the immune microenvironment and the desmoplastic stroma in OC metastasis (GSE137237). (**A**) Comparison of the predicted proportion of stromal cells in primary metastatic tumors from ovarian cancer patients. (**B**) Box plots depicting the expression levels of the 4 fibroblast functional modules (FFM1–4) in primary tumors and metastases. (**C**) Heatmap showing the Spearman correlation of 4 fibroblast functional modules and immune-associated markers in metastatic tissues. Red or blue boxes indicate positive or negative correlation between the immune-associated marker and the corresponding fibroblast functional modules (FFM1–4), respectively. Grey boxes indicate no correlation. Immune-activating or immune-suppressing markers are depicted, respectively, by the red or blue bars on the right. (**D**) A correlation analysis between the stromal cell types in the metastasis. The color scale for *R*-values are provided on the right. The sizes of the dots are also proportional to the magnitude of the *R*-value. *** *p* < 0.001, ** *p <* 0.01, * *p* < 0.05.

**Figure 3 cancers-12-03184-f003:**
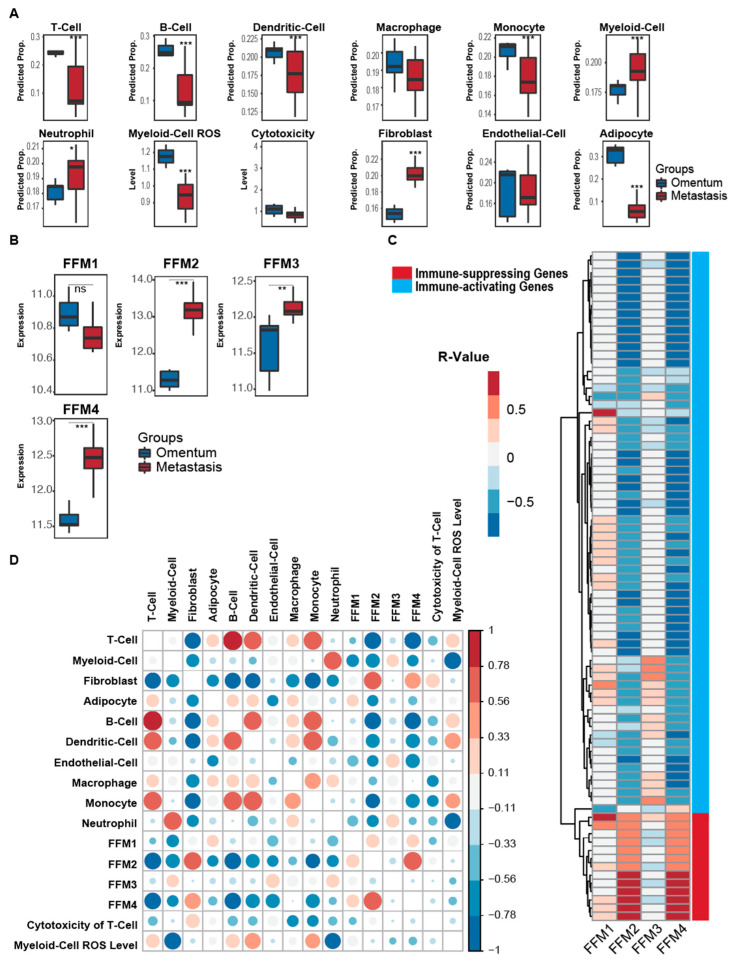
Comparison of the stromal cells of omental metastasis and normal omentum in an ovarian cancer mouse model. (**A**) Plots depicting predicted proportions of stromal cells in the normal omentum compared to omental metastases. (**B**) Box plots showing the expression levels of the 4 fibroblast functional modules (FFM1–4) in normal omental tissues compared to omental metastases. (**C**) Heatmap showing the Spearman correlation of 4 fibroblast functional modules and immune-associated markers in metastatic tissues. Red or blue boxes indicate positive or negative correlation between the immune-associated marker and the corresponding fibroblast functional module (FFM1–4), respectively. Grey boxes indicate no correlation. Red and blue bars on the right indicate the immune-suppressing and immune-activating markers, respectively. FFM, fibroblast functional module. (**D**) A correlation analysis between the stromal cell types in the metastasis. The color scale for *R*-values are provided on the right. The sizes of the dots are also proportional to the magnitude of the *R*-value. *** *p* < 0.001, ** *p* < 0.01, * *p <* 0.05.

**Figure 4 cancers-12-03184-f004:**
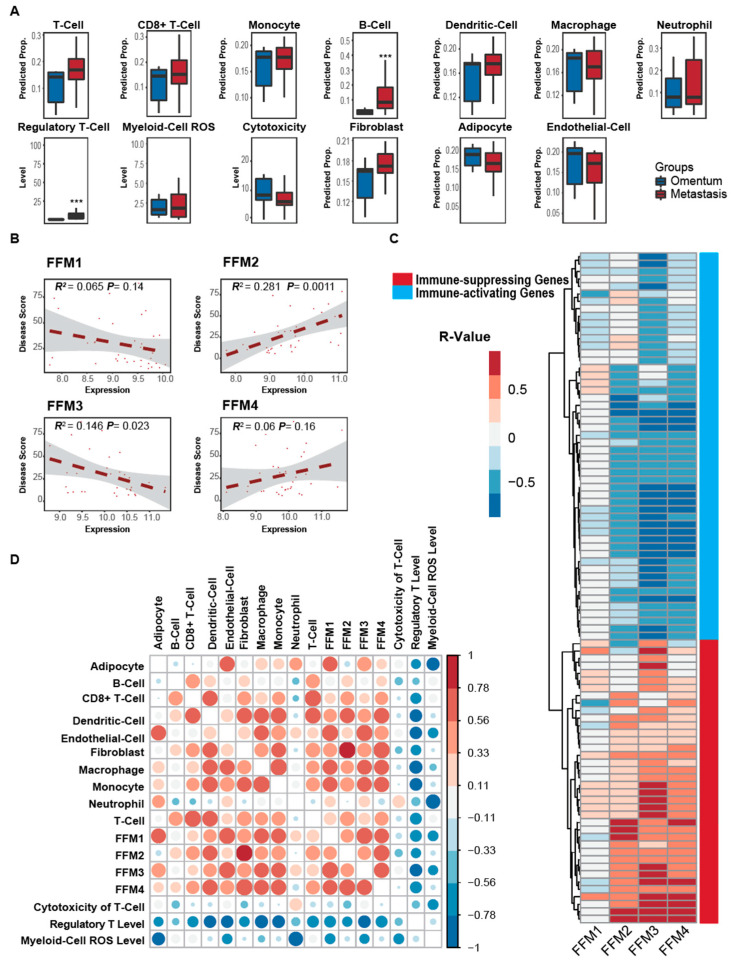
Analysis of stromal cellular composition of omental metastasis in human ovarian cancer patients with varying extent of disease spread in the omentum (GSE71340). (**A**) Comparison of the proportion of stromal cell types present in omentum with minimal metastases (omentum) and omentum with significant metastases (metastasis). (**B**) Scatter plots depicting the correlation between the expression of the 4 fibroblast functional modules (FFM1–4) with the extent of disease in the omentum specimen (disease score). (**C**) Heatmap showing the Spearman correlation of 4 fibroblast functional modules and immune-associated markers in tissues. Red or blue boxes indicate positive or negative correlation between the immune-associated marker and the corresponding fibroblast functional module (FFM1–4), respectively. Grey boxes indicate no correlation. Red and blue bars on the right depict the immune-suppressing and immune-activating markers, respectively. (**D**) A correlation analysis between the stromal cell types in the metastasis. The color scale for *R*-values are provided on the right. The sizes of the dots are also proportional to the magnitude of the *R*-value. *** *p* < 0.001.

**Figure 5 cancers-12-03184-f005:**
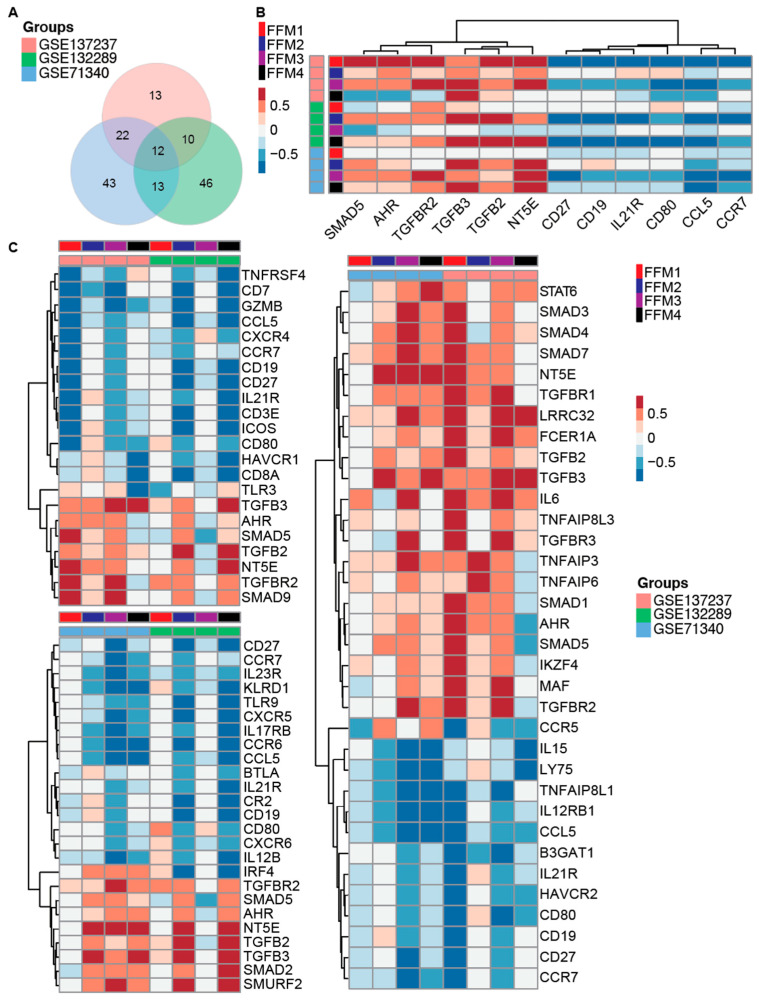
Immuno-markers that overlap between the primary tumors vs. metastasis (GSE137237), mouse omentum vs. omental metastasis (GSE132289) and human omentum vs. omental metastasis (GSE71340). (**A**) Venn diagram depicting the number of overlapping immune markers in GSE71340, GSE132289 and GSE137237 datasets. (**B**) Heatmap showing the Spearman correlation of 4 fibroblast functional modules and 12 immune-associated markers common to all 3 datasets in metastatic tissues. (**C**) Left top, heatmap showing the Spearman correlation of 4 fibroblast functional modules and 22 immune-associated markers that overlapped between GSE137237 and GSE132289. Left bottom, heatmap showing the Spearman correlation of 4 fibroblast functional modules and 25 immune-associated markers overlapped by GSE71340 and GSE132289 datasets. Right, heatmap showing the Spearman correlation of 4 fibroblast functional modules and 34 immune-associated markers overlapped by GSE71340 and GSE137237 datasets. Red or blue boxes in heatmap indicate positive or negative correlation between the immune-associated marker and the corresponding fibroblast functional module, respectively (FFM1–4).

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
