# Peer review of "Modulation of Immune Infiltration of Ovarian Cancer Tumor Microenvironment by Specific Subpopulations of Fibroblasts"

_cancers, 2020, doi:10.3390/cancers12113184_

Round 1
Reviewer 1 Report
As previously mentioned, since the authors use a semi-supervised approach to identify data set specific gene markers of cell sub-types, the reviewer feels that such approach should be further validated in this paper using other common data sets, such as the TCGA data set from ovarian cancer samples, and compare the cell identification data with expression subtypes (The 4 molecular subtypes were subsequently validated by TCGA network and termed: mesenchymal (MES), immunoreactive (IMM), differentiated (DIF), and proliferative (PRO)). Unfortunately, the authors only presented a table (Table 2) with co-expression of genes within the same FFM without any statistical insight on those correlations. This is rather useless for the reader as it is now. Furthermore, what is more important is to look at the expression of those genes within the different subtypes of ovarian cancer. In other words, where do these FFM classifications fall in different expression subtypes. The reviewer feel there needs to be a sort of validation for the existence of such FFMs outside of the authors analysis.
Moreover, as these FFMs could just well be artificial compartmentalizations of the expression data, the single cell analysis now performed could be a great insight into these FFM (Supp Fig 1). However, now the authors decided to only provide an average of expression of the markers by FFM, which only means that indeed they have been looking at fibroblasts from the beginning. What would be critical for the conclusions of this paper would be to find if they can actually identify sub populations of fibroblasts that could reproduce their FFM from the five patients analyzed with scRNA-seq.
The other minor concerns have been properly addressed.
Reviewer 2 Report
Authors have revised and no more comments.
Author Response
Reviewer did not have any additional comments.
Reviewer 3 Report
The authors responded nicely to the points raised by the reviewer.
Author Response
Reviewer did not have any additional comments.
This manuscript is a resubmission of an earlier submission. The following is a list of the peer review reports and author responses from that submission.
Round 1
Reviewer 1 Report
The authors used a bioinformatic approach to understand the role of cancer associated fibroblasts (CAFs) in determining the tumor immune microenvironment and potentially predicting efficacy of immunotherapies. They claim they identified four functional modules of CAFs in ovarian cancer that are associated with the tumor microenvironment and metastasis of ovarian cancer. They also found a immune suppressive function of COL1, COL3 and COL5 expressing CAFs, which the authors claim are part of the fibroblast functional module 2 (FFM2) in primary ovarian cancer and omentum metastases. This type of analysis can potentially help understand the tumor promoting immune microenvironment, with implications toward the effectiveness of ovarian cancer immunotherapies.
Since the authors use a semi-supervised approach to identify data set specific gene markers of cell sub-types that is most likely under revision now (available on Biorxv), the reviewer feels that such approach should be further validated in this paper using other common data sets, such as the TCGA data set from ovarian cancer samples, and compare the cell identification data with expression subtypes (mesenchymal, etc..). In other words, where do these FFM fall in different expression subtypes. The reviewer feel there needs to be a sort of validation for the existence of such FFMs outside of the authors analysis. These FFMs could just well be artificial compartmentalizations of the expression data. Maybe by comparing these FFM to the published HGSC subtypes would be an interesting way to understand these different subtypes.
Also, it would be important to look at expression of the collagens in tumor samples (from IHC for example) and related those findings to the FFM, especially in the context of COL IV, which is mostly part of the basal membrane, and maybe mostly expressed not by fibroblasts but by epithelial or endothelial cells. The reviewer’s major concern is that these functional modules may not be that functional after all, and further dissection of their significance should be, at least discussed.
Minor issues:
Supp Fig 1A: PCA plot should also show scree plot to see how much variance is explained in the other projections.
What is the meaning of the blue and brown circles surrounding the unidentified samples in the PCA plots?
Supp Fig 1B:“Gene ontology analysis of the differentially expressed genes”: between which samples? It is vaguely mentioned in the main text but has to be very clearly specified in the figure legend.
It is mentioned in the discussion section that: “Furthermore, we compared our immune cell type data with multiplexed immunohistochemistry immune profiling of HGSOC chemotherapy responders and poor responders [63]. Interestingly, increased T-cell populations in metastasis was consistent with increased T-cell infiltration observed in good responders to carbo/taxol chemotherapy.” Where is this analysis in the paper?
Reviewer 2 Report
In this study authors tried to define the role of CAF in ovarian tumor progression and metastasis by analyzing three different databases and identified four functional modules of CAFs in ovarian cancer associated with ovarian tumor metastasis. They further analyzed gene profiles in those four modules and identified pro and anti-inflammatory genes. This is an interesting and important study to understand how TME contributes to ovarian tumor metastasis. However, whole studies are based on bioinformatics analysis and lack of experimental validation such as gene expression by RT-PCR or immunostaining in primary or metastatic tumors.
Reviewer 3 Report
This is an intersting manuscript describing the role of tumor associated fibroblast cells in ovarian cancer metastasis.
However it is a pure data base analyses based on sets that contain limited number of samples: GSE132289 data includes 37 mouse samples (16 HGS, 3 GEM, 9 cl, 9 normal omental tissues), 8 human samples, GSE137237 data contains 11 pairs of matched primary and metastases tumor tissue samples, and GSE71340 data contains 35 samples
The analyses that has been done is staight forward and the results are interesting for the readership.
But the assumtion that findings of the study indicate that the FFM2 subtype, with high expression of collagen 1,3,5 plays a key role in tumor immune tolerance should be re-evaluated on a higher number of cases.
Is there an disadvantage for survival of ovarian cancer patients if FFM2 subtype cells with high expression of collagen 1,3,5 are present?
Could this be analysed on a higher number of patients?
